# Accelerating Neural Architecture Search using Performance Prediction

**Bowen Baker**[*], **Otkrist Gupta**,[*] **Ramesh Raskar**
Media Laboratory
Massachusetts Institute of Technology
Cambridge, MA 02142, USA
{bowen,otkrist,raskar}@mit.edu

**Nikhil Naik**
Faculty of Arts and Sciences
Harvard University
Cambridge, MA 02138, USA
naik@fas.harvard.edu

## Abstract

Methods for neural network hyperparameter optimization and architecture search are computationally expensive due to the need to train a large number of model configurations. In this paper, we show that simple regression models can predict the final performance of partially trained model configurations using features based on network architectures, hyperparameters, and time-series validation performance data. We empirically show that our performance prediction models are much more accurate than prominent Bayesian counterparts, are simpler to implement, and are faster to train. Our models can predict final performance in both visual classification and language modeling domains, are effective for predicting performance of drastically varying model architectures, and can even generalize between model classes. Using these prediction models, we also implement an early stopping method for hyperparameter optimization and architecture search, which obtains a speedup of a factor up to 6x in both hyperparameter optimization and architecture search. Finally, we empirically show that our early stopping method can be seamlessly incorporated into both reinforcement learning-based architecture selection algorithms and bandit based search methods. Through extensive experimentation, we empirically show our performance prediction models and early stopping algorithm are state-of-the-art in terms of prediction accuracy and speedup achieved while still identifying the optimal model configurations.

## 1 Introduction

Significant human expertise and labor is required for designing high-performing neural network architectures and successfully training them for different applications. Ongoing research in two areas—architecture search and hyperparameter optimization—attempts to reduce the amount of human intervention required for these tasks. Hyperparameter optimization methods (e.g., Hutter et al. (2011); Snoek et al. (2015); Li et al. (2017)) focus primarily on obtaining good optimization hyperparameter configurations for training human-designed networks, whereas architecture search algorithms (Bergstra et al., 2013; Verbancsics & Harguess, 2013; Baker et al., 2017; Zoph & Le, 2017) aim to design neural network architectures from scratch. Both sets of algorithms require training a large number of neural network configurations for identifying the right set of hyperparameters or the right network architecture and are hence computationally expensive.

When sampling many different model configurations, it is likely that many subpar configurations will be explored. Human experts are quite adept at recognizing and terminating suboptimal model configurations by inspecting their partial learning curves. In this paper we seek to emulate this behavior and automatically identify and terminate subpar model configurations in order to speedup both architecture search and hyperparameter optimization methods. Our method parameterizes learning curve trajectories with simple features derived from model architectures, training hyperparameters, and early time-series measurements from the learning curve. We use these features to train a set of simple regression models that predict the final validation performance of partially trained neural network configurations using a small training set of fully trained curves, and we empirically validate our

---

[*]Equal Contribution

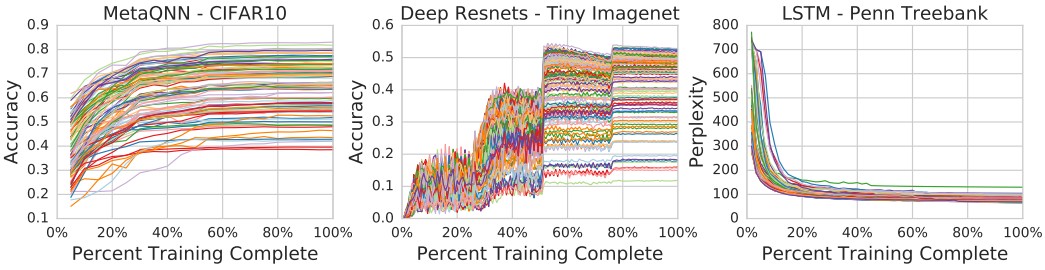

Figure 1: **Example Learning Curves:** Example learning curves from experiments considered in this paper. Note the diversity in convergence times and overall learning curve shapes.

method in both image classification and language modeling domains. We use these predictions along with uncertainty estimates obtained from small model ensembles to construct a simple early stopping algorithm that can speedup both architecture search and hyperparameter optimization methods.

While there is some prior work on neural network performance prediction using Bayesian methods (Domhan et al., 2015; Klein et al., 2017), our proposed method is significantly more accurate, accessible, and efficient. We hope that our work leads to inclusion of neural network performance prediction and early stopping in the practical neural network training pipeline.

## 2 RELATED WORK

**Neural Network Performance Prediction:** There has been limited work on predicting neural network performance during training. Domhan et al. (2015) introduce a weighted probabilistic model for learning curves and utilize this model for speeding up hyperparameter search in small convolutional neural networks (CNNs) and fully-connected networks (FCNs). Building on Domhan et al. (2015), Klein et al. (2017) train Bayesian neural networks for predicting unobserved learning curves using a training set of fully and partially observed learning curves. Both methods rely on expensive Markov chain Monte Carlo (MCMC) sampling procedures and handcrafted learning curve basis functions. We also note that Swersky et al. (2014) develop a Gaussian Process kernel for predicting individual learning curves, which they use to automatically stop and restart configurations.

**Architecture Search:** We define architecture search as an algorithmic approach for designing neural network architectures from scratch. The earliest architecture search approaches were based on genetic algorithms (Schaffer et al., 1992; Stanley & Miikkulainen, 2002; Verbancsics & Harguess, 2013) or Bayesian optimization (Bergstra et al., 2013; Shahriari et al., 2016). More recently, reinforcement learning methods have become popular. Baker et al. (2017) use Q-learning to design competitive CNNs for image classification. Zoph & Le (2017) use policy gradients to design state-of-the-art CNNs and recurrent cell architectures. Several methods for architecture search (Cortes et al., 2017; Negrinho & Gordon, 2017; Zoph et al., 2017; Brock et al., 2017; Suganuma et al., 2017) have been proposed this year since the publication of Baker et al. (2017) and Zoph & Le (2017).

**Hyperparameter Optimization:** We define hyperparameter optimization as an algorithmic approach for finding optimal values of architecture independent hyperparameters such as learning rate and batch size, along with a limited search through the network design space. Bayesian hyperparameter optimization methods include those based on sequential model-based optimization (SMAC) (Hutter et al., 2011), Gaussian processes (GP) (Snoek et al., 2012), TPE (Bergstra et al., 2013), and neural networks Snoek et al. (2015). However, random search or grid search is most commonly used in practical settings (Bergstra & Bengio, 2012). Recently, Li et al. (2017) introduced Hyperband, a multi-armed bandit-based efficient random search technique that outperforms state-of-the-art Bayesian optimization methods.

## 3 NEURAL NETWORK PERFORMANCE PREDICTION

We first describe our model for neural network performance prediction, followed by a description of the datasets used to evaluate our model, and finally present experimental results.

### 3.1 Modeling Learning Curves

Our goal is to model the validation performance $y_T$ of a neural network configuration $\mathbf{x} \in \mathcal{X} \subset \mathbb{R}^d$ at epoch $T \in \mathbb{Z}^+$ using previous performance observations $y(t)$. For each configuration $\mathbf{x}$ trained for $T$ epochs, we record a time-series $y(T) = y_1, y_2, \ldots, y_T$ of validation performances. We train a population of $n$ configurations, obtaining a set $\mathcal{S} = \{(\mathbf{x}^1, y^1(T)), (\mathbf{x}^2, y^2(T)), \ldots, (\mathbf{x}^n, y^n(T))\}$. This problem formulation is very similar to Klein et al. (2017). Note that most architecture and hyperparameter search methods naturally collect $\mathcal{S}$.

We propose to use a set of features $u_{\mathbf{x}}$, derived from the neural network configuration $\mathbf{x}$, along with a subset of time-series performances $y(\tau) = (y_t)_{t=1,2,\ldots,\tau}$ (where $1 \leq \tau < T$) from $\mathcal{S}$ to train a regression model for estimating $y_T$. We use models without natural support for variable size input, so we train $T-1$ regression models, where each successive model uses one more point of the time-series validation data. As we shall see in subsequent sections, this use of *sequential regression models* (SRM) is more computationally efficient and more precise than methods that train a single model.

**Features:** We use features based on time-series (TS) validation performances, architecture parameters (AP), and hyperparameters (HP). (1) TS: Assume we are training the $\tau^{\text{th}}$ model in the SRM. TS features include the validation performances $y(\tau) = (y_t)_{t=1,2,\ldots,\tau}$, the first-order differences of validation performances, i.e. $y_t' = (y_t - y_{t-1})$, the second-order differences of validation performances, i.e. $y_t'' = (y_t' - y_{t-1}')$, and mean and standard deviation of performances. (2) AP: These include total number of weights and number of layers. (3) HP: These include all hyperparameters used for training the neural networks, e.g. initial learning rate and learning rate decay (full list in Appendix Table 4).

### 3.2 Datasets and Training Procedures

Figure 1 shows example learning curves from three of the datasets considered in our experiments. We provide brief summary of the datasets below. Please see Appendix Section A for further details on the search space, preprocessing, hyperparameters and training settings of all datasets.

Datasets with Varying Architectures:

**Deep Resnets (TinyImageNet):** We sample 500 ResNet architectures and train them on the TinyImageNet* dataset (containing 200 classes with 500 training images of $32 \times 32$ pixels) for 140 epochs. We vary depths, filter sizes and number of convolutional filter block outputs. The network depths vary between 14 and 110.

**Deep Resnets (CIFAR-10):** We sample 500 39-layer ResNet architectures from a search space similar to Zoph & Le (2017), varying kernel width, kernel height, and number of kernels. We train these models for 50 epochs on CIFAR-10.

**MetaQNN CNNs (CIFAR-10 and SVHN):** We sample 1,000 model architectures from the search space detailed by Baker et al. (2017), which allows for varying the numbers and orderings of convolution, pooling, and fully connected layers. The models are between 1 and 12 layers for the SVHN experiment and between 1 and 18 layers for the CIFAR-10 experiment. Each architecture is trained on SVHN and CIFAR-10 datasets for 20 epochs.

**LSTM (PTB):** We sample 300 LSTM models and train them on the Penn Treebank dataset for 60 epochs, evaluating perplexity on the validation set. We vary number of LSTM cells and hidden layer inputs between 10-1400.

Datasets with Varying Hyperparameters:

**Cuda-Convnet (CIFAR-10 and SVHN):** We train Cuda-Convnet architecture (Krizhevsky, 2012) with varying values of initial learning rate, learning rate reduction step size, weight decay for convolutional and fully connected layers, and scale and power of local response normalization layers. We train models with CIFAR-10 for 60 epochs and with SVHN for 12 epochs.

---

*https://tiny-imagenet.herokuapp.com/

| Dataset | $\nu$-SVR | OLS | BLR | RF |
|---|---|---|---|---|
| MetaQNN (CIFAR-10) | $97.22 \pm 0.31$ | $97.04 \pm 0.42$ | $97.11 \pm 0.32$ | $94.50 \pm 0.78$ |
| Resnet (TinyImageNet) | $87.12 \pm 4.65$ | $88.41 \pm 3.76$ | $82.40 \pm 1.65$ | $86.39 \pm 3.58$ |
| LSTM (Penn Treebank) | $98.03 \pm 1.46$ | $87.83 \pm 21.53$ | $97.99 \pm 1.90$ | $57.28 \pm 24.98$ |

Table 1: **Performance Prediction Model Comparison:** We report the coefficient of determination $R^2 * 100$ for four standard methods. Each model is trained with 100 samples on 25% of the learning curve. We find that $\nu$-SVR works best on average, though not by a large margin.

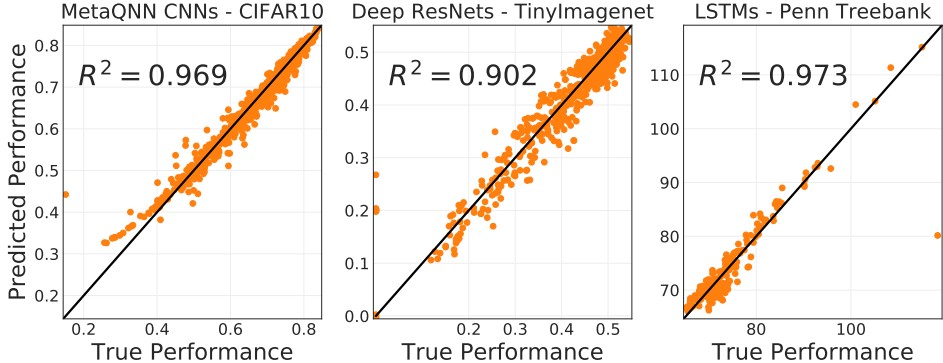

Figure 2: **Predicted vs True Values of Final Performance:** We show the shape of the predictive distribution on three experiments: MetaQNN models, Deep Resnets, and LSTMs. Each $\nu$-SVR model is trained with 100 configurations with data from 25% of the learning curve. We predict validation set classification accuracy for MetaQNN and Deep ResNets, and perplexity for LSTMs.

## 3.3 PREDICTION PERFORMANCE

**Choice of Regression Method:** We now describe our results for predicting final neural network performance. For all experiments, we train our SRMs on 100 randomly sampled neural network configurations. We obtain the best performing method using random hyperparameter search over 3-fold cross-validation. We then compute the regression performance over the remainder of the dataset using the coefficient of determination $R^2$. We repeat each experiment 10 times and report the results with standard errors. We experiment with a few simple regression models, including ordinary least squares (OLS), random forests (RF), Bayesian linear regression (BLR), and $\nu$-support vector machine regression ($\nu$-SVR). For both OLS and $\nu$-SVR, we choose between the linear and RBF kernels based on prediction performance—after hyperparameter search, in case of $\nu$-SVR. We chose 100 as the number of sampled configurations somewhat arbitrarily, though we found that prediction performance was quite poor below 50 and improves steadily after. As seen in Table 1, $\nu$-SVR performs the best on most datasets, though not by a large margin. For the rest of this paper, we use $\nu$-SVR unless otherwise specified.

**Ablation Study on Feature Sets:** In Table 2, we compare the predictive ability of different feature sets, training SVR with time-series (TS) features obtained from 25% of the learning curve, along with features of architecture parameters (AP), and hyperparameters (HP). TS features explain the largest fraction of the variance in all cases and found that in general AP are more important that HP. AP features almost match TS on the ResNet (TinyImageNet) dataset, indicating that choice of architecture has a large influence on accuracy for ResNets. Figure 2 shows the true vs. predicted performance for all test points in three datasets, trained with TS, AP, and HP features.

**Generalization To Out Of Distribution Configurations:** We also test to see whether our SRMs can accurately predict the performance of out-of-distribution neural networks. For instance, if we train a $\nu$-SVR model on the Deep Resnet (TinyImagenet) dataset at 25% of the learning curve observed on configurations below median depth (60 layers), we can achieve $R^2 = 0.86$ on configurations above median depth. The corresponding number for prediction below median depth is 0.89. We repeated this experiment all datasets with different hyperparameter values below/above the median and found that SRMs generally showed consistent performance across such splits. Full results can be found in Appendix Section H.

| Feature Set | MetaQNN (CIFAR-10) | ResNets (TinyImageNet) | LSTM (Penn Treebank) | Cuda-Convnet (CIFAR-10) |
|---|---|---|---|---|
| HP | $-2.96 \pm 2.71$ | $0.48 \pm 7.24$ | $-1.72 \pm 1.62$ | $0.48 \pm 7.24$ |
| AP | $15.38 \pm 4.66$ | $67.66 \pm 9.08$ | $13.76 \pm 12.08$ | $67.66 \pm 9.08$ |
| TS | $97.00 \pm 0.52$ | $82.71 \pm 4.31$ | $97.38 \pm 1.49$ | $82.71 \pm 4.31$ |
| AP+HP | $14.89 \pm 5.12$ | $70.25 \pm 6.42$ | $13.76 \pm 12.08$ | $70.25 \pm 6.42$ |
| TS+HP | $97.02 \pm 0.52$ | $82.67 \pm 6.81$ | $97.38 \pm 1.49$ | $82.67 \pm 6.81$ |
| TS+AP | $97.12 \pm 0.37$ | $87.36 \pm 4.23$ | $97.87 \pm 1.68$ | $87.36 \pm 4.23$ |
| TS+AP+HP | $97.15 \pm 0.48$ | $86.05 \pm 8.24$ | $97.87 \pm 1.68$ | $86.05 \pm 8.24$ |

Table 2: **Ablation Study on Feature Sets:** We report the $R^2 * 100$ metric for different combinations of features. Time-series features (TS) refers to the partially observed learning curves, architecture parameters (AP) refer to the number of layers and number of weights in a deep model, and hyperparameters (HP) refer to the optimization parameters such as learning rate. All results are with $\nu$-SVR. 25% of learning curve used for models including TS features.

### 3.3.1 Comparison with Existing Methods:

We now compare the neural network performance prediction ability of SRMs with three existing learning curve prediction methods: (1) Bayesian Neural Network (BNN) (Klein et al., 2017), (2) the learning curve extrapolation (LCE) method (Domhan et al., 2015), and (3) the last seen value (LastSeenValue) heuristic (Li et al., 2017). When training the BNN, we not only present it with the subset of fully observed learning curves but also all other partially observed learning curves from the training set. While we do not present the partially observed curves to the SRMs for training, we felt this was a fair comparison as an SRM uses the entire partially observed learning curve during inference. Methods (2) and (3) do not incorporate prior learning curves during training. Figure 3 shows the $R^2$ obtained by each method for predicting the final performance versus the percent of the learning curve used for training the model. We see that in all neural network configuration spaces and across all datasets, $\nu$-SVR outperform the competing methods. Bayesian linear regression (BLR) also performs comparably well to $\nu$-SVR and better than the more complex Bayesian methods (BNN and LCE). The LastSeenValue heuristic only becomes viable when the configurations are near convergence, and its performance is worse than an SRM for very deep models. We also find that the SRMs outperform the LCE method in all experiments, even after we remove a few extreme prediction outliers produced by LCE. Finally, while BNN outperforms the LastSeenValue and LCE methods when only a few iterations have been observed, it does worse than our proposed method. In summary, we show that our simple SRMs outperforms existing Bayesian approaches on predicting neural network performance on modern, very deep models in computer vision and language modeling tasks.

Since most of our experiments perform stepwise learning rate decay; it is conceivable that the performance gap between SRMs and both LCE and BNN results from a lack of sharp jumps in their basis functions. For completeness, we also experimented with exponential learning rate decay (ELRD), which the basis functions in LCE are designed for. We trained 630 random nets with ELRD, from the 1000 MetaQNN-CIFAR10 nets. Predicting from 25% of the learning curve, the $R^2$ is 0.95 for $\nu$-SVR, 0.48 for LCE (with extreme outlier removal, negative without), and 0.31 for BNN. This comparison illuminates another benefit of our method: we do not require handcrafted basis functions to model new learning curve types.

Moreover, SRMs are faster to train and do inference in than LCE and BNN. On 1 core of a Intel 6700k CPU, an $\nu$-SVR with 100 training points trains in 0.006 seconds, and each inference takes 0.00006 seconds. In comparison, the LCE code takes 60 seconds and BNN code takes 0.024 seconds on the same hardware for each inference.

## 4 Applying Performance Prediction For Early Stopping

To speed up hyperparameter optimization and architecture search methods, we develop an algorithm to determine whether to continue training a partially trained model configuration using our sequential regression models. This approach follows the idea proposed by Domhan et al. (2015) and further expanded on by Klein et al. (2017). Concretely, if we would like to sample $N$ total neural network

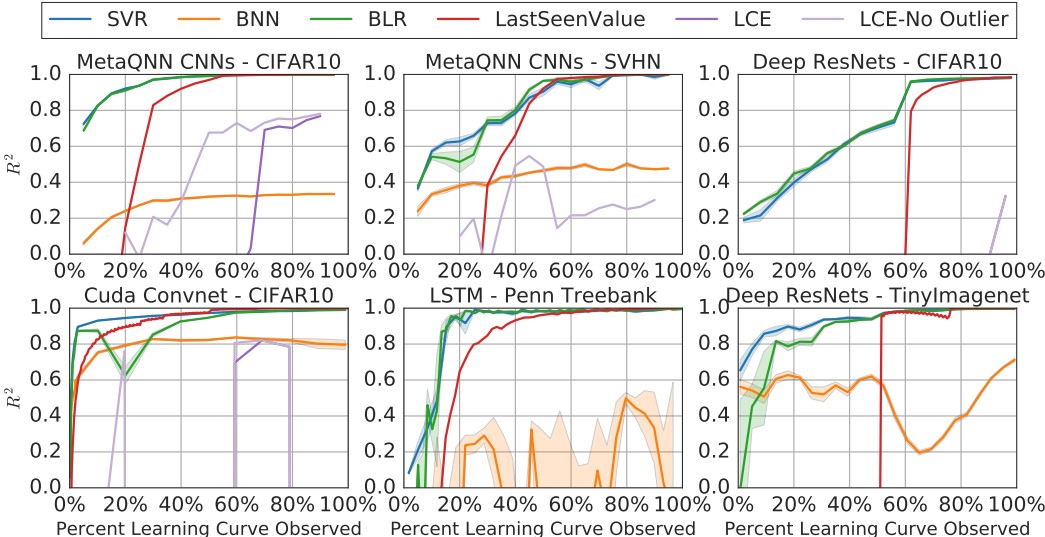

Figure 3: **Performance Prediction Results:** We plot the performance of each method versus the percent of learning curve observed. For BNN and $\nu$-SVR, we sample 10 different training sets, plot the mean $R^2$, and shade the corresponding standard error. We compare our method against BNN (Klein et al., 2017), LCE (Domhan et al., 2015), and a "last seen value" heuristic (Li et al., 2017). Absent results for a model indicate that it did not achieve a positive $R^2$. The results for Cuda-Convnet on the SVHN dataset are shown in Appendix Figure 11.

configurations, we begin by sampling and training $n \ll N$ configurations to create a training set $\mathcal{S}$. We then train a model $f(x_f)$, where in our case $f$ is an SRM with $T-1$ individual models, to predict $y_T$. Now, given the current best performance observed $y_{\text{BEST}}$, we would like to terminate training a new configuration $\mathbf{x}_{\text{NEW}}$ at iteration $\tau$ given its partial learning curve $y_{\text{NEW}}(\tau)$ if $f(x_{\text{NEW}f}) = \hat{y}_T \leq y_{\text{BEST}}$ so as to not waste computational resources exploring a suboptimal configuration.

However, in the case $f$ has poor out-of-sample generalization, we may mistakenly terminate the optimal configuration. If we assume that our estimate can be modeled as a Gaussian perturbation of the true value $\hat{y}_T \sim \mathcal{N}(y_T, \sigma(\mathbf{x}, \tau))$, then we can find the probability $p(\hat{y}_T \leq y_{\text{BEST}} | \sigma(\mathbf{x}, \tau)) = \Phi(y_{\text{BEST}}; y_T, \sigma)$, where $\Phi(\cdot; \mu, \sigma)$ is the CDF of $\mathcal{N}(\mu, \sigma)$. Note that in general the uncertainty will depend on both the configuration and $\tau$, the number of points observed from the learning curve. Because frequentist models do not admit a natural estimate of uncertainty, we assume that $\sigma$ is independent of $\mathbf{x}$ yet still dependent on $\tau$ and estimate it via Leave One Out Cross Validation (LOOCV). In addition, we show results using a Bayesian linear regression SRM, which has natural uncertainty estimates, removing the need for LOOCV.

Now that we can estimate the model uncertainty, given a new configuration $\mathbf{x}_{\text{NEW}}$ and an observed learning curve $y_{\text{NEW}}(\tau)$, we may set our termination criteria to be $p(\hat{y}_T \leq y_{\text{BEST}}) \geq \Delta$. $\Delta$ balances the trade-off between increased speedups and risk of prematurely terminating good configurations. In many cases, one may want several configurations that are close to optimal, for the purpose of ensembling. We offer two modifications in this case. First, one may relax the termination criterion to $p(\hat{y}_T \leq y_{\text{BEST}} - \delta) \geq \Delta$, which will allow configurations within $\delta$ of optimal performance to complete training. One can alternatively set the criterion based on the $n^{\text{th}}$ best configuration observed, guaranteeing that with high probability the top $n$ configurations will be fully trained.

### 4.1 EARLY STOPPING FOR ARCHITECTURE SEARCH

Baker et al. (2017) train a $Q$-learning agent to design convolutional neural networks. In this method, the agent samples architectures from a large, finite space by traversing a path from input layer to termination layer. However, the MetaQNN method uses 100 GPU-days to train 2700 neural architectures and the similar experiment by Zoph & Le (2017) utilized 10,000 GPU-days to train 12,800 models on CIFAR-10. The amount of computing resources required for these approaches

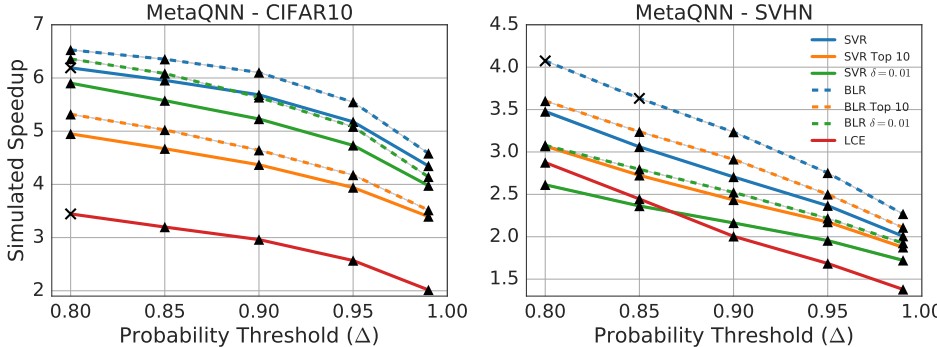

Figure 4: **Simulated Speedup in MetaQNN Search Space:** We compare the three variants of the early stopping algorithm presented in Section 4 for both a $\nu$-SVR and BLR SRM. Each SRM is trained using the first 100 learning curves, and each algorithm is tested on 10 independent orderings of the model configurations. Triangles indicate an algorithm that successfully recovered the optimal model for more than half of the 10 orderings, and X's indicate those that did not.

makes them prohibitively expensive for large datasets (e.g., Imagenet) and larger search spaces. The main computational expense of reinforcement learning-based architecture search methods is training the neural network configuration to $T$ epochs (where $T$ is typically a large number at which the network stabilizes to peak accuracy).

We now detail the performance of $\nu$-SVR and BLR SRMs in speeding up architecture search. First, we take 1,000 random models from the MetaQNN (Baker et al., 2017) search space. We simulate the MetaQNN algorithm by taking 10 random orderings of each set and running our early stopping algorithm. We compare against the LCE early stopping algorithm (Domhan et al., 2015) as a baseline, which has a similar probability threshold termination criterion. Our SRM trains off of the first 100 fully observed curves, while the LCE model trains from each individual partial curve and can begin early termination immediately. Despite this "burn in" time needed by an SRM, it is still able to outperform the LCE model (Figure 4). In addition, fitting the LCE model to a learning curve takes between 1-3 minutes on a modern CPU due to expensive MCMC sampling, and it is necessary to fit a new LCE model each time a new point on the learning curve is observed. However, the difference in computing time between the methods is not as important as the difference in prediction accuracies.

We furthermore simulate early stopping for ResNets trained on CIFAR-10. We found that only the probability threshold $\Delta = 0.99$ resulted in recovering the top model consistently. However, even with such a conservative threshold, the search was sped up by a factor of 3.4 over the baseline. While we do not have the computational resources to run the full experiment from Zoph & Le (2017), our method could provide similar gains in large scale architecture searches.

It is not enough, however, to simply simulate the speedup because architecture search algorithms typically use the observed performance in order to update an acquisition function to inform future sampling. In the reinforcement learning setting, the performance is given to the agent as a reward, so we also empirically verify that substituting $\hat{y}_T$ for $y_T$ does not cause the MetaQNN agent to converge to a subpar policy. Replicating the MetaQNN experiment on CIFAR-10 (see Figure 5), we find that integrating early stopping with the $Q$-learning procedure does not disrupt learning and resulted in a speedup of 3.8x with $\Delta = 0.99$. After training the top models to 300 epochs, we also

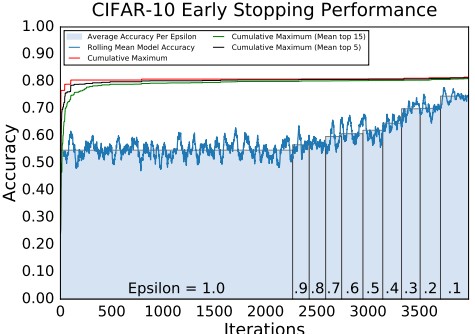

Figure 5: **MetaQNN on CIFAR-10 with Early Stopping:** A full run of the MetaQNN algorithm (Baker et al., 2017) on the CIFAR-10 dataset with early stopping. We use the $\nu$-SVR SRM with a probability threshold $\Delta = 0.99$. Light blue bars indicate the average model accuracy per decrease in $\epsilon$, which represents the shift to a more greedy policy. We also plot the cumulative best, top 5, and top 15 performance to show that the agent continues to find better architectures.

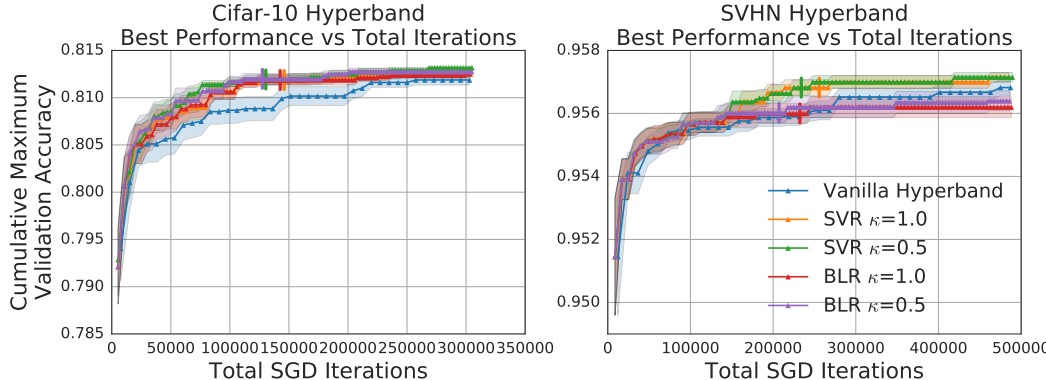

Figure 6: **Simulated Max Accuracy vs SGD Iterations for Hyperband:** We show the trajectories of the maximum performance so far versus total computational resources used for 40 consecutive Hyperband runs with $\eta = 4.0$ and $\Delta = 0.95$. Small vertical lines indicate the point at which each f-Hyperband run has searched over the same number of models as vanilla Hyperband. On Cifar-10, both BLR and $\nu$-SVR f-Hyperbands outperform vanilla Hyperband, but on SVHN only $\nu$-SVR f-Hyperbands do. Each triangle marks the completion of full Hyperband algorithm.

find that the resulting performance (just under 93%) is on par with original results of Baker et al. (2017).

## 4.2 EARLY STOPPING
### FOR HYPERPARAMETER OPTIMIZATION

Recently, Li et al. (2017) introduced Hyperband, a random search technique based on multi-armed bandits that obtains state-of-the-art performance in hyperparameter optimization in a variety of settings. The Hyperband algorithm trains a population of models with different hyperparameter configurations and iteratively discards models below a certain percentile in performance among the population until the computational budget is exhausted or satisfactory results are obtained. Among previous work, Klein et al. (2017) have improved Hyperband by algorithmically choosing the next models to evaluate.

### 4.2.1 FAST HYPERBAND

We present a Fast Hyperband (f-Hyperband) algorithm based on our early stopping scheme. During each iteration of successive halving, Hyperband trains $n_i$ configurations to $r_i$ epochs. In f-Hyperband, we train an SRM to predict $y_{r_i}$ and do early stopping within each iteration of successive halving. We initialize f-Hyperband in exactly the same way as vanilla Hyperband, except once we have trained 100 models to $r_i$ iterations, we begin early stopping for all future successive halving iterations that train to $r_i$ iterations. By doing this, we exhibit no computational overhead over Hyperband other than training the SRMs, which we have shown to be cheap. We also introduce a parameter $\kappa$ which denotes the proportion of the $n_i$ models in each iteration that must be trained to the full $r_i$ iterations. This is similar to setting the criterion based on the $n^{\text{th}}$ best model in the previous section. See Appendix section C for the full algorithmic representation of f-Hyperband.

We empirically evaluate f-Hyperband using Cuda-Convnet trained on CIFAR-10 and SVHN datasets. Figure 6 shows that f-Hyperband evaluates the same number of unique configurations as Hyperband within half the compute time, while achieving the same final accuracy within standard error. Moreover, when we run f-Hyperband for the same amount of time as Hyperband, f-Hyperband with a $\nu$-SVR SRM outperforms f-Hyperband. When reinitializing hyperparameter searches, one can use previously-trained set of SRMs to achieve even larger speedups. Figure 12 in Appendix shows that one can safely achieve up to a 4x speedup in such cases.

## 5 CONCLUSION

In this paper we introduce a simple, fast, and accurate model for predicting future neural network performance using features derived from network architectures, hyperparameters, and time-series performance data. We show that the performance of drastically different network architectures can be jointly learned and predicted on both image classification and language models. Using our simple algorithm, we can speedup hyperparameter search techniques with complex acquisition functions, such as a $Q$-learning agent, by a factor of 3x to 6x and Hyperband—a state-of-the-art hyperparameter search method—by a factor of 2x, without disturbing the search procedure. We outperform all competing methods for performance prediction in terms of accuracy, train and test time, and speedups obtained on hyperparameter search methods. We hope that the simplicity and success of our method will allow it to be easily incorporated into current hyperparameter optimization pipelines for deep neural networks. With the advent of large scale automated architecture search (Baker et al., 2017; Zoph & Le, 2017), methods such as ours will be vital in exploring even larger and more complex search spaces.

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

APPENDIX

## A  DATASETS AND ARCHITECTURES

**Deep Resnets (TinyImageNet):** We sample 500 ResNet architectures and train them on the TinyImageNet[†] dataset (containing 200 classes with 500 training images of $32 \times 32$ pixels) for 140 epochs. We vary depths, filter sizes and number of convolutional filter block outputs. Filter sizes are sampled from $\{3, 5, 7\}$ and number of filters is sampled from $\{2, 3, 4, ..., 22\}$. Each ResNet block is composed of three convolutional layers followed by batch normalization and summation layers. We vary the number of blocks from 2 to 18, giving us networks with depths varying between 14 and 110. Each network is trained for 140 epochs, using Nesterov optimizer. The learning rate is set to 0.1 and learning rate reduction and momentum are set to 0.1 and 0.9 respectively.

**Deep Resnets (CIFAR-10):** We sample 500 39-layer ResNet architectures from a search space similar to Zoph & Le (2017), varying kernel width, kernel height, and number of kernels. We train these models for 50 epochs on CIFAR-10. Each architecture consists of 39 layers: 12 *conv*, a 2x2 *max pool*, 9 *conv*, a 2x2 *max pool*, 15 *conv*, and *softmax*. Each *conv* layer is followed by batch normalization and a ReLU nonlinearity. Each block of 3 *conv* layers are densely connected via residual connections and also share the same kernel width, kernel height, and number of learnable kernels. Kernel height and width are independently sampled from $\{1, 3, 5, 7\}$ and number of kernels is sampled from $\{6, 12, 24, 36\}$. Finally, we randomly sample residual connections between each block of *conv* layers. Each network is trained for 50 epochs using the RMSProp optimizer, with weight decay $10^{-4}$, initial learning rate 0.001, and a learning rate reduction to $10^{-5}$ at epoch 30 on the CIFAR-10 dataset.

**MetaQNN CNNs (CIFAR-10 and SVHN):** We sample 1,000 model architectures from the search space detailed by Baker et al. (2017), which allows for varying the numbers and orderings of convolution, pooling, and fully connected layers. The models are between 1 and 12 layers for the SVHN experiment and between 1 and 18 layers for the CIFAR-10 experiment. Each architecture is trained on SVHN and CIFAR-10 datasets for 20 epochs. Table 3 displays the state space of the MetaQNN algorithm.

| Layer Type | Layer Parameters | Parameter Values |
|---|---|---|
| Convolution (C) | $i \sim$ Layer depth
$f \sim$ Receptive field size
$\ell \sim$ Stride
$d \sim$ # receptive fields
$n \sim$ Representation size | $< 12$
Square. $\in \{1, 3, 5\}$
Square. Always equal to 1
$\in \{64, 128, 256, 512\}$
$\in \{(\infty, 8], (8, 4], (4, 1]\}$ |
| Pooling (P) | $i \sim$ Layer depth
$(f, \ell) \sim$ (Receptive field size, Strides)
$n \sim$ Representation size | $< 12$
Square. $\in \{(5, 3), (3, 2), (2, 2)\}$
$\in \{(\infty, 8], (8, 4]$ and $(4, 1]\}$ |
| Fully Connected (FC) | $i \sim$ Layer depth
$n \sim$ # consecutive FC layers
$d \sim$ # neurons | $< 12$
$< 3$
$\in \{512, 256, 128\}$ |
| Termination State | $s \sim$ Previous State
$t \sim$ Type | 
Global Avg. Pooling/Softmax |

Table 3: **Experimental State Space For MetaQNN.** For each layer type, we list the relevant parameters and the values each parameter is allowed to take. The networks are sampled beginning from the starting layer. Convolutional layers are allowed to transition to any other layer. Pooling layers are allowed to transition to any layer other than pooling layers. Fully connected layers are only allowed to transition to fully connected or softmax layers. A convolutional or pooling layer may only go to a fully connected layer if the current image representation size is below 8. We use this space to both randomly sample and simulate the behavior of a MetaQNN run as well as directly run the MetaQNN with early stopping.

**LSTM (PTB):** We sample 300 LSTM models and train them on the Penn Treebank dataset for 60 epochs. Number of hidden layer inputs and lstm cells was varied from 10 to 1400 in steps of 20. Each network was trained for 60 epochs with batch size of 50 and trained the models using stochastic

---

[†]https://tiny-imagenet.herokuapp.com/

gradient descent. Dropout ratio of 0.5 was used to prevent overfitting. Dictionary size of 400 words was used to generate embeddings when vectorizing the data.

**Cuda-Convnet (CIFAR-10 and SVHN):** We train Cuda-Convnet architecture (Krizhevsky, 2012) with varying values of initial learning rate, learning rate reduction step size, weight decay for convolutional and fully connected layers, and scale and power of local response normalization layers. We train models with CIFAR-10 for 60 epochs and with SVHN for 12 epochs. Table 4 show the hyperparameter ranges for the Cuda Convnet experiments.

| Experiment | Hyperparameter | Scale | Min | Max |
|---|---|---|---|---|
| CIFAR-10, Imagenet, SVHN | Initial Learning Rate | Log | $5 \times 10^{-5}$ | 5 |
| | Learning Rate Reductions | Integer | 0 | 3 |
| CIFAR-10, SVHN | Conv1 $L_2$ Penalty | Log | $5 \times 10^{-5}$ | 5 |
| | Conv2 $L_2$ Penalty | Log | $5 \times 10^{-5}$ | 5 |
| | Conv3 $L_2$ Penalty | Log | $5 \times 10^{-5}$ | 5 |
| | FC4 $L_2$ Penalty | Log | $5 \times 10^{-5}$ | 5 |
| | Response Normalization Scale | Log | $5 \times 10^{-6}$ | 5 |
| | Response Normalization Power | Linear | $1 \times 10^{-2}$ | 3 |

Table 4: Range of hyperparameter settings used for the Hyperband experiment (Section 4.1)

## B Hyperparameter selection in Performance Prediction Models

When training performance prediction models we divided the data into training and validation and used 3-fold cross validation to select optimal hyperparameters. The models were then trained on full training data using the best hyperparameters. For random forests we varied number of trees between 10 and 800, and varied ratio of number of features from 0.1 to 0.5. For $\nu$-SVR, we perform a random search over 1000 hyperparameter configurations from the space $C \sim \text{LogUniform}(10^{-5}, 10)$, $\nu \sim \text{Uniform}(0, 1)$, and $\gamma \sim \text{LogUniform}(10^{-5}, 10)$ (when using the RBF kernel). For Bayesian linear regression, we perform random search over the hyperparameters of the gamma prior distribution over both noise and weights in the range $[10^{-7}, 10^{-5}]$.

## C f-Hyperband

Algorithm 1 of this text replicates Algorithm 1 from Li et al. (2017), except we initialize two dictionaries: $D$ to store training data and $M$ to store performance prediction models. $D[r]$ will correspond to a dictionary containing all datasets with prediction target epoch $r$. $D[r][\tau]$ will correspond to the dataset for predicting $y_r$ based on the observed $y(t)_{1-\tau}$, and $M[r][\tau]$ will hold the corresponding performance prediction model. We will assume that the performance prediction model will have a `train` function, and a `predict` function that will return the prediction and standard deviation of the prediction. In addition to the standard Hyperband hyperparameters $R$ and $\eta$, we include $\Delta$ and $\delta$ described in Section 4 and $\kappa$. During each iteration of successive halving, we train $n_i$ configurations to $r_i$ epochs; $\kappa$ denotes the fraction of the top $n_i$ models that should be run to the full $r_i$ iterations. This is similar to setting the criterion based on the $n^{\text{th}}$ best model in the previous section.

We also detail the `run_then_return_validation_loss` function in Algorithm 2. This algorithm runs a set of configurations, adds training data from observed learning curves, trains the performance prediction models when there is enough training data present, and then uses the models to terminate poor configurations. It assumes we have a function `max_k`, which returns the $k^{\text{th}}$ max value or $-\infty$ if the list has less than $k$ values.

---

**Algorithm 1:** f-Hyperband

---

**input**   : $R$    –    (Max resources allocated to any configuration)

              $\eta$    –    (default $\eta = 3$)

              $\Delta$    –    (Probability threshold for early termination)

              $\delta$    –    (Performance offset for early termination)

              $d$    –    (# points required to train performance predictors)

              $\kappa$    –    (Proportion of models to train)

**initialize :** $D = \text{dict}()$

              $M = \text{dict}()$

              $s_{\max} = \lfloor \log_\eta(R) \rfloor$

              $B = (s_{\max} + 1)R$

1   **for** $s \in \{s_{max}, \ldots, 0\}$ **do**

2      $n = \lceil \frac{B}{R} \frac{\eta^s}{s+1} \rceil, \quad r = R\eta^{-s}$

3      `// begin SUCCESSIVEHALVING with (n, r) inner loop`

4      $T = $ `get_hyperparameter_configuration(n)`

5      **for** $i \in \{0, \ldots, s\}$ **do**

6          $n_i = \lfloor n\eta^{-i} \rfloor, \quad r_i = r\eta^i$

7          $n_{\text{next}} = \lfloor \frac{n_i}{\eta} \rfloor$ **if** $i! = s$ **else** 1

8          $L = $ `run_then_return_validation_loss`$(T, r_i, n_{\text{next}}, D, M)$

9          $T = $ `top_k`$(T, L, \lfloor \frac{n_i}{\eta} \rfloor)$

10      **end**

11 **end**

---

**Algorithm 2:** `run_then_return_validation_loss`

---

**input**   : $T$    –    hyperparameter configurations

              $r$    –    resources to use for training

              $n$    –    # configurations in next iteration of successive halving

              $D$    –    dictionary storing training data

              $M$    –    dictionary storing performance prediction models

**initialize :** $L = $ `[ ]`

1   **for** $t \in T$ **do**

2      $\ell = $ `[ ]`

3      **for** $i \in \{0, \ldots, r - 1\}$ **do**

4          $\ell_i = $ `run_one_epoch_return_validation_loss`$(t)$

5          $\ell$`.append`$(\ell_i)$

6          **if** $M[r][i]$`.trained()` **then**

7              $\hat{y}_r, \sigma = M[r][i]$`.predict`$(\ell)$

8              **if** $\Phi($`max_k`$(L,\ \kappa n) - \delta; \hat{y}_r, \sigma) \geq \Delta$ **then**

9                  $L$`.append`$(\hat{y}_r)$

10                  `break`

11              **end**

12          **end**

13          **else if** $i == r - 1$ **then**

14              $L$`.append`$(\ell_i)$

15          **end**

16      **end**

17      **if** `length`$(D[r][0]) < d$ **and** `length`$(\ell) == r$ **then**

18          $\{D[r][i]$`.append`$(\{\ell[0, \ldots, i], \ell[r]\}) : i \in \{0, \ldots, r - 1\}\}$

19          **if** $\text{not} M[r][i]$`.trained()` **then**

20              $M[r][i]$`.train`$(D[r][i])$

21          **end**

22      **end**

23 **end**

24 **return** $L$

---

# D   F-HYPERBAND WITH SVR ACQUISITION FUNCTION

Similar to Klein et al. (2017), it is possible to use our simple models as an acquisition function for Hyperband. As shown in Table 2, our models have reasonable performance with only AP and HP features used, meaning we can use these models to rank hyperparameter settings before training the configurations. In particular, for each Hyperband iteration, we randomly select 10,000 configurations, weight them by their predicted performance and sample from this weighted distribution. In Figure 7, we compare f-Hyperband with this $\nu$-SVR acquisition function to f-Hyperband with the standard uniform acquisition function and also to vanilla Hyperband. It seems that the acquisition function could help slightly, but it does not improve performance across all experiments, and where it does improve performance it does so only slightly.

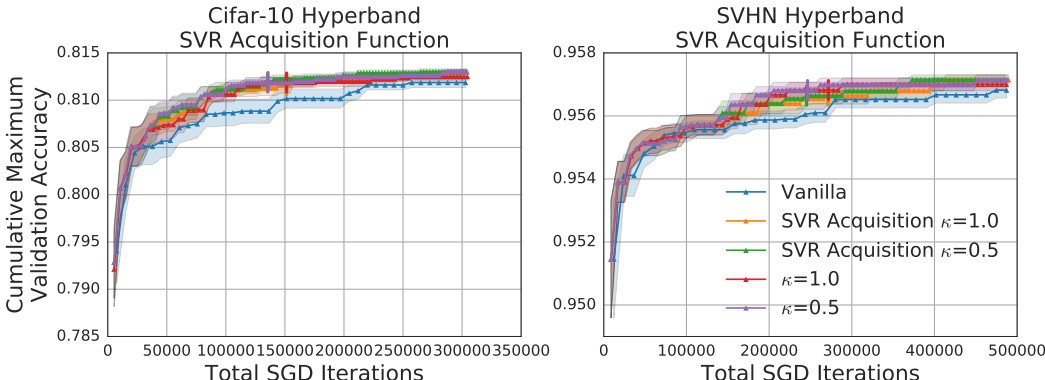

Figure 7: **f-Hyperband with SVR Acquisition Function:** We compare f-Hyperband with an SVR acquisition function to f-Hyperband with the uniform acquisition function (this is the standard version described in the main text) and to vanilla Hyperband. The results shown are inconclusive; it seems that there could be some benefit of incorporating the SVR acquisition function, but it isn't significant enough to make a strong statement.

# E   GAUSSIAN ERROR ASSUMPTION

In order to have uncertainty estimates with frequentist models, we estimated the mean and variance of the error from leave-one-out cross validation and assumed that the error was Gaussian. In Figure 8 we compare example error distributions between training and validation from a $\nu$-SVR SRM. Visually one can see that the assumption holds very well for both visual classification tasks and holds slightly less well for the language modeling task. In Figure 9, we show the mean log likelihood of held out prediction errors being drawn from a Gaussian parameterized by the mean and variance of the error on the training set is very close to the mean log likelihood of samples drawn from the same distribution, which shows that our Gaussian error assumption is relatively strong.

# F   ANALYZING IMPORTANCE OF FEATURES IN THE PREDICTION MODEL

We used a linear $\nu$-SVR model to compare the weights of all features, which are normalized before training. Figure 10 shows the weight trends for time-series features, and Tables 5, 6, and 7 compare weights of some architecture features and hyperparameter features to statistics of time-series features for the MetaQNN (CIFAR-10), LSTMs (Penn Treebank), and Deep Resnets (TinyImagenet) experiments, respectively. We found the following main insights across datasets:

- The time-series (TS) features are on average have higher weights than HP and AP features (which is confirmed by our ablation studies in Table 2 as well). The original validation accuracies ($y_t$) have higher weights on average than the first-order differences ($y_t{}'$) and second-order differences ($y_t{}''$).

- In general, later epochs in $y_t$ have higher weights. In other words, the latest performance of the model available for prediction is much more important for performance prediction than initial

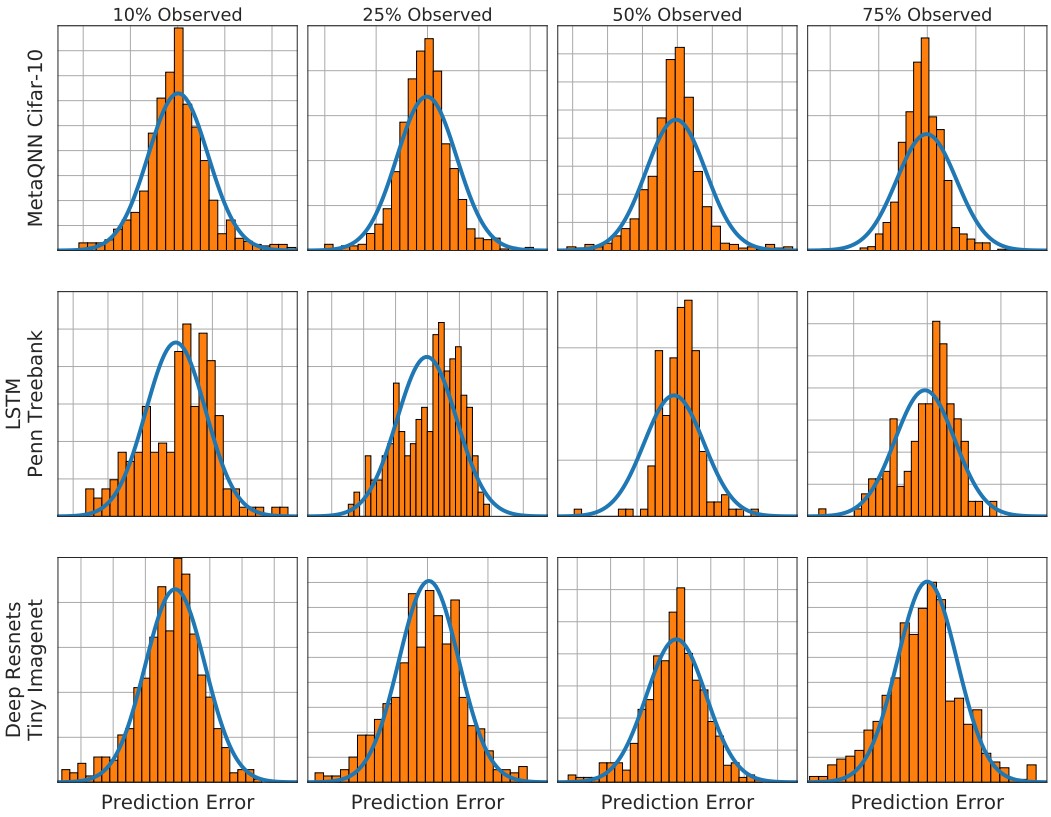

Figure 8: **Example Error Distributions:** Here we compare example error distributions between training and validation from a $\nu$-SVR SRM. From left to right we see plots for models trained with increasing amounts of the learning curve. The blue line shows the Gaussian probability density function estimated from the training set with LOOCV. The orange histogram shows the error distribution of the held out validation set.

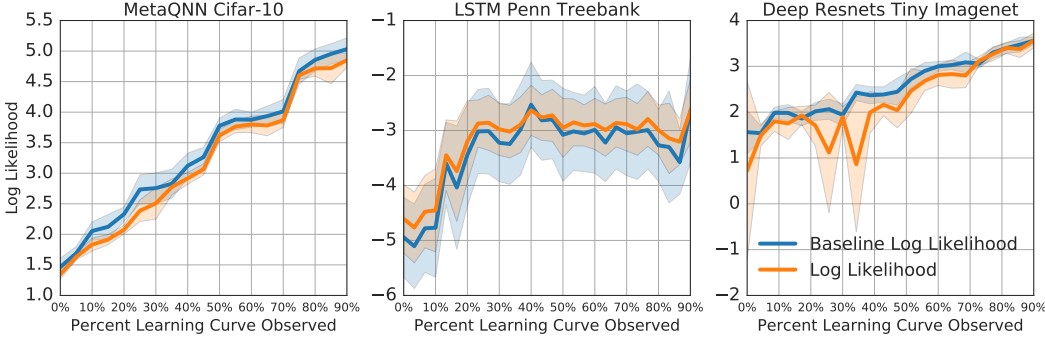

Figure 9: **Error Log Likelihood:** In orange we show the mean log likelihood of prediction errors measured from validation sets being drawn from the Gaussian distribution parameterized by the mean and variance of the training error, again measured with LOOCV. For a baseline, we show the mean log likelihood for 1,000,000 samples drawn from the same Gaussian. We calculate these metrics over 10 data splits and report the mean and standard error.

performance, when the model has just started training. However, the weights for earlier performance metrics are non-zero, indicating there are high order autoregressive functions being learned.

- Early values of $(y_t')$ also have high weights, which indicates learning quickly in the beginning of training is a predictor of better final performance in our datasets.

- Among AP and HP features, the total number of parameters and depth have non-zero weight for the CNNs, and they are assigned weights comparable or higher than late epoch accuracies ($y_t$). However, they have much lower weight in the LSTM experiment. The number of filters in each convolutional layer also has a high positive weight for CNNs. In general, architectural features are much more important for CNNs as compared to LSTMs. Hyperparameters like initial learning rate and step size were generally not as important as the architectural features, which is corroborated by the ablation study in Table 1 in the main text.

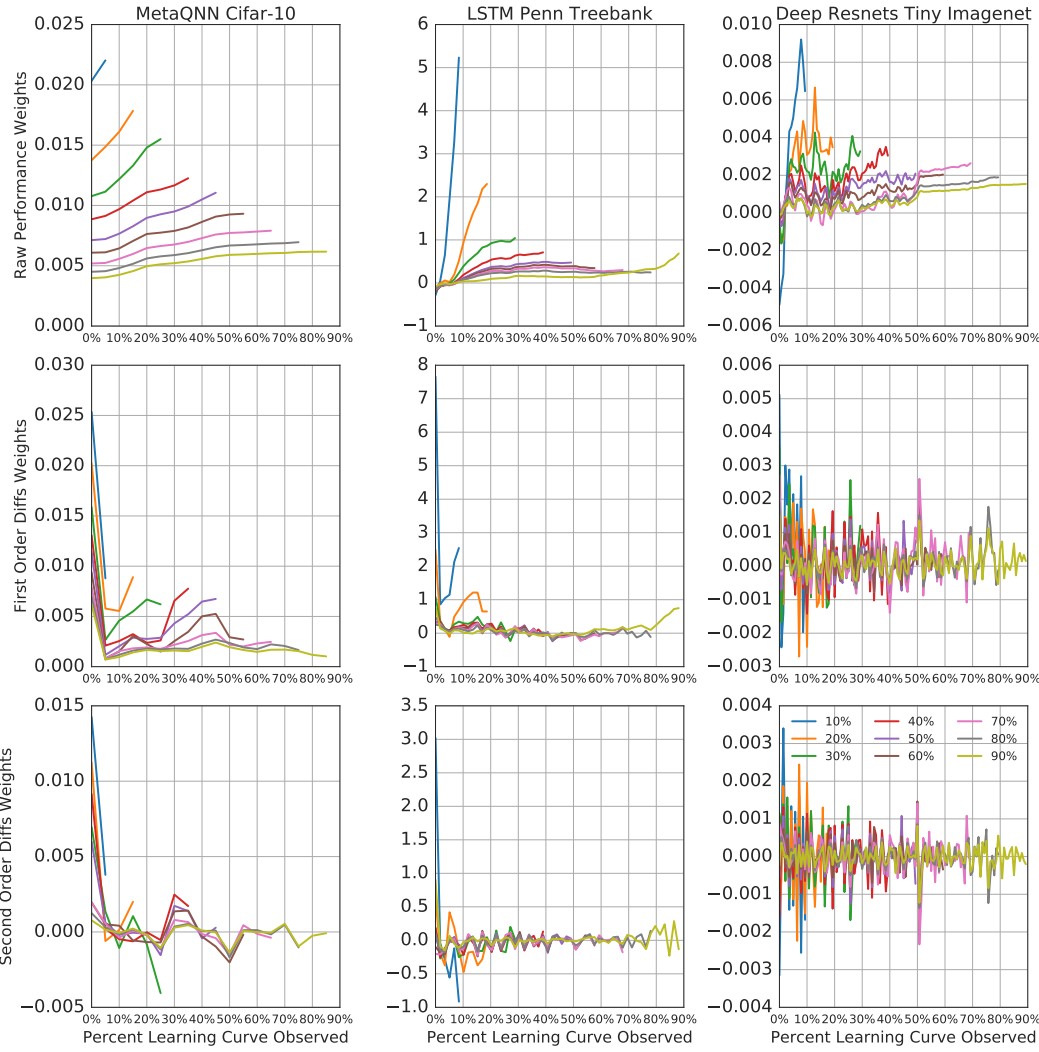

Figure 10: **Time Series Weights:** We show the trends for weights for time-series features, along with first- and second-order differences.

| Percent Observed | Mean Perf | Std Dev Perf | Mean 1st Order Diff | Mean 2nd Order Diff | Depth | # Params | Learning Rate |
|---|---|---|---|---|---|---|---|
| 10 | 0.022989 | 0.013845 | 0.017069 | 0.009021 | 0.014638 | 0.005541 | 0.005355 |
| 20 | 0.016971 | 0.010882 | 0.010117 | 0.003494 | 0.008153 | 0.003408 | 0.005335 |
| 30 | 0.013733 | 0.007396 | 0.006921 | 0.002550 | 0.004199 | 0.002098 | 0.003310 |
| 40 | 0.011125 | 0.007023 | 0.005031 | 0.001903 | 0.002847 | 0.003041 | 0.002968 |
| 50 | 0.009409 | 0.005836 | 0.004594 | 0.001234 | 0.000578 | 0.000223 | 0.001107 |
| 60 | 0.008099 | 0.004768 | 0.003370 | 0.000923 | 0.001306 | -0.000906 | 0.000287 |
| 70 | 0.007016 | 0.004246 | 0.002573 | 0.000627 | 0.000458 | -0.000777 | -0.000324 |
| 80 | 0.006190 | 0.003417 | 0.002159 | 0.000494 | 0.000454 | 0.000355 | 0.000583 |
| 90 | 0.005553 | 0.003141 | 0.001792 | 0.000371 | 0.000382 | 0.000020 | 0.000259 |

Table 5: **Summary of weights for a linear kernel $\nu$-SVR SRM trained on configurations from MetaQNN (CIFAR-10).**

| Percent Observed | Mean Perf | Std Dev Perf | Mean 1st Order Diff | Mean 2nd Order Diff | Depth | # Params | Learning Rate Step |
|---|---|---|---|---|---|---|---|
| 10 | 1.129178 | -0.952712 | 2.562584 | 0.822748 | -2.608709 | 0.599052 | -1.940918 |
| 20 | 0.388479 | -0.168798 | 0.828343 | 0.287950 | -1.148619 | 0.390338 | -0.667870 |
| 30 | 0.202656 | -0.254058 | 0.282163 | 0.132048 | -1.134044 | -0.033193 | -0.301467 |
| 40 | 0.149271 | -0.205868 | 0.183959 | 0.093839 | -0.280646 | 0.050759 | -0.057207 |
| 50 | 0.129166 | -0.160091 | 0.127230 | 0.078865 | -0.219122 | -0.044931 | -0.119674 |
| 60 | 0.116550 | -0.160410 | 0.122557 | 0.085949 | -0.128821 | 0.040022 | -0.097907 |
| 70 | 0.112243 | -0.144278 | 0.104209 | 0.078584 | -0.082918 | -0.077808 | -0.057439 |
| 80 | 0.102105 | -0.127418 | 0.078328 | 0.068932 | -0.062506 | 0.054454 | -0.006713 |
| 90 | 0.106653 | -0.067369 | 0.141887 | 0.074515 | 0.079686 | -0.169534 | 0.158086 |

Table 6: **Summary of weights for a linear kernel $\nu$-SVR SRM trained on configurations from LSTMs (Penn Treebank).**

| Percent Observed | Mean Perf | Std Dev Perf | Mean 1st Order Diff | Mean 2nd Order Diff | Depth | # Params | Kernel Size |
|---|---|---|---|---|---|---|---|
| 10 | 0.008486 | 0.008576 | 0.001861 | 0.001502 | 0.005985 | 0.010987 | 0.000267 |
| 20 | 0.004723 | 0.004851 | 0.001092 | 0.000996 | 0.001709 | 0.005239 | 0.002484 |
| 30 | 0.003229 | 0.002848 | 0.000789 | 0.000619 | 0.004533 | 0.002553 | -0.001138 |
| 40 | 0.002394 | 0.002985 | 0.000660 | 0.000557 | 0.003443 | 0.001092 | -0.000302 |
| 50 | 0.001754 | 0.001993 | 0.000461 | 0.000396 | 0.001318 | 0.001081 | 0.000072 |
| 60 | 0.001456 | 0.001946 | 0.000370 | 0.000309 | 0.000968 | 0.000559 | 0.000036 |
| 70 | 0.001428 | 0.002678 | 0.000499 | 0.000344 | 0.001161 | -0.000590 | -0.000971 |
| 80 | 0.001168 | 0.001895 | 0.000352 | 0.000256 | 0.000455 | 0.000131 | -0.001092 |
| 90 | 0.001015 | 0.001640 | 0.000285 | 0.000215 | 0.000208 | -0.000091 | -0.000958 |

Table 7: **Summary of weights for a linear kernel $\nu$-SVR SRM trained on configurations from Deep Resnets (TinyImagenet).**

## G   ADDITIONAL PLOTS

Figure 11 shows the performance prediction results for the Cuda Convnet SVHN experiment. The instability of BLR with low percentage of learning curve observed may be what caused suboptimal performance of BLR f-Hyperband shown in Figure 6. Figure 12 shows the relative speedup of f-Hyperband over vanilla Hyperband for each consecutive iteration of Hyperband. However, Figure 13, which shows results for f-Hyperband with pretrained $\nu$-SVR SRMs, shows that more aggressive settings of f-Hyperband result in suboptimal performance. We see that we can safely get a 3-4x speedup over vanilla Hyperband with pretrained SRMs.

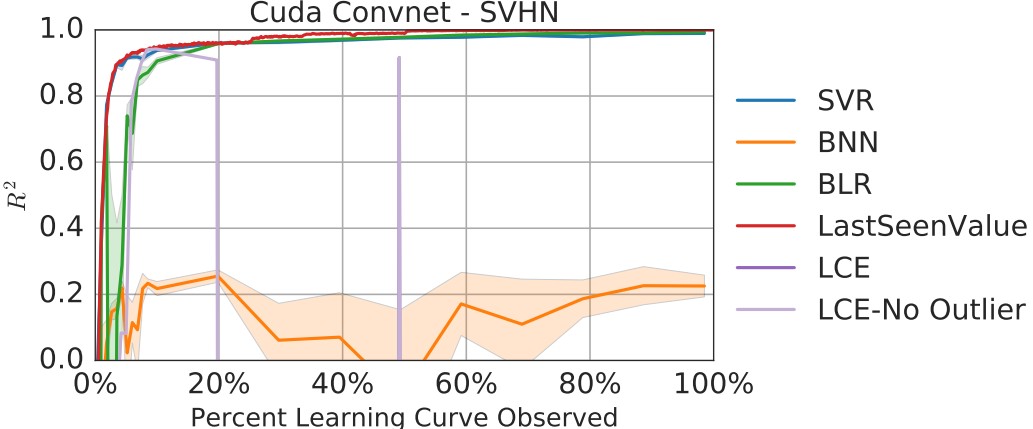

Figure 11: **Cuda Convnet SVHN Performance Prediction Results:** We plot the performance of each method versus the percent of learning curve observed for the Cuda Convnet SVHN experiment. For BNN, $\nu$-SVR, and BLR we sample 10 different training sets, plot the mean $R^2$, and shade the corresponding standard error. We compare our method against BNN (Klein et al., 2017), LCE (Domhan et al., 2015), and a "last seen value" heuristic (Li et al., 2017). Absent results for a model indicate that it did not achieve a positive $R^2$.

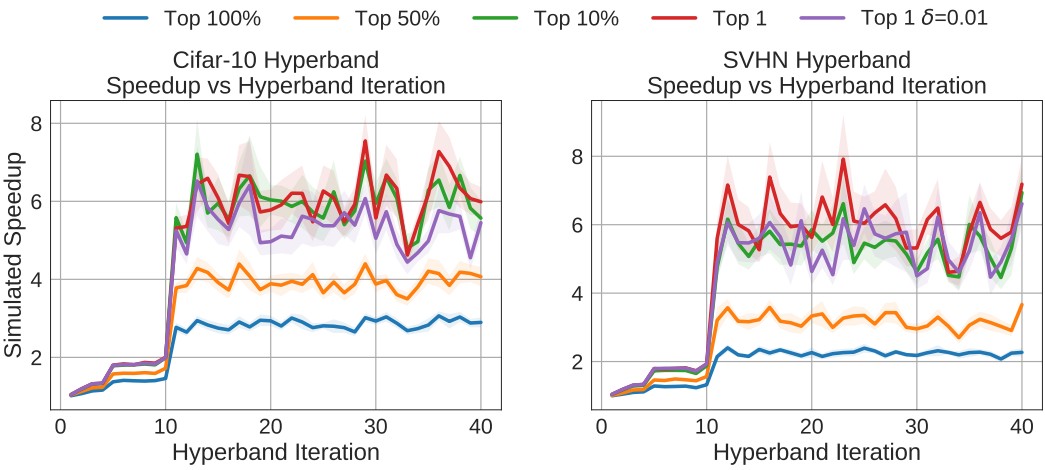

Figure 12: **Simulated Speedup on Hyperband vs Hyperband Iteration:** We show the speedup using the f-Hyperband algorithm over Hyperband on 40 consecutive runs with $\eta = 4.0$ and $\Delta = 0.95$. The major jump in speedup comes at iteration 10, where we have trained more than 100 models to the full $R$ iterations.

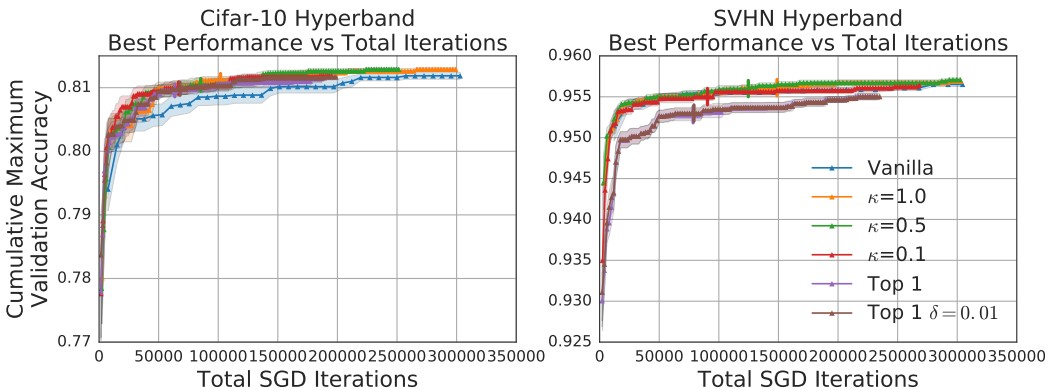

Figure 13: **Simulated f-Hyperband with Pretrained SRMs:** Each f-Hyperband model is initialized with a pretrained SRM, which simulates the case where one reinitializes an experiment. Every triangle indicates a full run of Hyperband, and small vertical lines indicate the point at which the f-Hyperband runs have completed the same number of full Hyperband runs as shown for vanilla Hyperband. Top 1 indicates only training the best model to $r_i$ iterations, and $\delta$ represents an offset off the threshold used for early stopping (See Section 4).More aggressive f-Hyperband settings lead to suboptimal performance, indicating that there is a limit to how much speedup f-Hyperband can achieve.

## H SRM ROBUSTNESS TO OUT OF DISTRIBUTION CONFIGURATIONS

Table 8 shows that SRMs are relatively robust when training below or above the median of a specific architecture or training hyperparameter and testing on the other split.

| **Cuda Convnet (CIFAR-10)** | Learning Rate | WeightDecay | LRN Scale | LRN Power |
|---|---|---|---|---|
| Test < Median | $93.06 \pm 0.18$ | $95.93 \pm 0.21$ | $97.27 \pm 0.0068$ | $96.96 \pm 0.13$ |
| Test > Median | $81.97 \pm 1.26$ | $96.19 \pm 0.13$ | $93.37 \pm 0.23$ | $95.05 \pm 0.19$ |

| **LSTMs (Penn Treebank)** | Number of Nodes | Depth |
|---|---|---|
| Test < Median | $98.08 \pm 0.26$ | $98.58 \pm 0.04$ |
| Test > Median | $22.83 \pm 5.55$ | $96.96 \pm 0.91$ |

| **Deep Resnets (TinyImagenet)** | Depth | Number of Kernels | Kernel Size |
|---|---|---|---|
| Test < Median | $89.49 \pm 1.84$ | $85.12 \pm 2.24$ | $81.83 \pm 0.59$ |
| Test > Median | $85.92 \pm 1.78$ | $62.37 \pm 0.82$ | $89.61 \pm 0.31$ |

| **MetaQNN (CIFAR-10)** | Depth |
|---|---|
| Test < Median | $96.69 \pm 0.16$ |
| Test > Median | $79.31 \pm 5.66$ |

Table 8: Accuracy when splitting test and train set based on median values for hyperparameters for different datasets.

