# OpenReview forum: "Accelerating Neural Architecture Search using Performance Prediction"
_ICLR.cc/2018/Conference — Invite to Workshop Track_

### Official Review · AnonReviewer3 · 2017-11-26
**Some Interesting Results, but Unsatisfying Analysis**

**Rating:** 6
**Confidence:** 4

**Review:**


This paper explores the use of simple models for predicting the final
validation performance of a neural network, from intermediate values
during training.  It uses support vector regression to show that a
relatively small number of samples of hyperparameters, architectures,
and validation time series can lead to reasonable predictions of
eventual performance.  The paper performs a modest evaluation of such
simple models and shows surprisingly good r-squared values.  The
resulting simple prediction framework is then used for early stopping,
in particular within the Hyperband hyperparameter search algorithm.

There's a lot that I like about this paper, in particular the ablation
study to examine which pieces matter, and the evaluation of a couple
of simple models.  Ultimately, however, I felt like the paper was
somewhat unsatisfying as it left open a large number of obvious
questions and comparisons:

- The use of the time series is the main novelty.  In the AP, HP and
  AP+HP cases of Table 2, it is essentially the same predictive setup
  of SMAC, BO, and other approaches that are trying to model the map
  from these choices to out-of-sample performance.  Doesn't the good
  performance without TS on, e.g., ResNets in Table 2 imply that the
  Deep ResNets subfigure in Figure 3 should start out at 80+?

- In light of the time series aspect being the main contribution, a
  really obvious question is: what does it learn about the time
  series?  The linear models do very well, which means it should be
  possible to look at the magnitude of the weights.  Are there any
  surprising long-range dependencies?  The fact that LastSeenValue
  doesn't do as well as a linear model on TS alone would seem to
  indicate that there are higher order autoregressive coefficients.
  That's surprising and the kind of thing that a scientific
  investigation here should try to uncover; it's a shame to just put
  up a table of numbers and not offer any analysis of why this works.

- In Table 1 the linear SVM uniformly outperforms the RBF SVM, so why
  use the RBF version?

- Given that the paper seeks to use uncertainty in estimates and the
  entire regression setup could be trivially made Bayesian with no
  significant computational cost over a kernelized SVM or OLS,
  especially if you're doing LOOCV to estimate uncertainty in the
  frequentist models.  Why not include Bayesian linear regression and
  Gaussian process regression as baselines?

- Since the model gets information from the AP and HP before doing any
  iterations, why not go on and use that to help select candidates?

- I don't understand how speedup is being computed in Figure 4.

- I'd like a more explicit accounting of whether 0.00006 seconds vs
  0.024 seconds is something we should care about in this kind of
  work, when the steps can take minutes or hours on a GPU.

- How useful is r-squared as a measure of performance in this setting?
  My experience has been that most of the search space has very poor
  performance and the objective is to find the small regions that work
  well.

Minor things:

- y' (prime) gets overloaded in Section 3.1 as a derivative and then
  in Section 4 as a partial learning curve.

- "... is more computationally and ..."

- "... our results for performing final ..."

---

> ### Author Response · Authors · 2018-01-04
> **Responses**
>
> Thank you for your thoughtful review! Please see below for responses to your questions.
>
> * “Doesn't the good performance without TS on, e.g., ResNets in Table 2 imply that the Deep ResNets subfigure in Figure 3 should start out at 80+?”
>
> - We investigated the difference between Table 2 and Figure 3 and we found that this difference was a result of the different hyperparameter ranges used to optimize the SVRs in the two experiments that we ran to compute results for Table 2 and Figure 3. To ensure that the results are consistent across all the experiment in the paper, we have now updated all results in the paper with SVR models that have the kernel as an additional hyperparameter searched over (with linear and RBF kernel as options). We have updated the results of Table 2 and Figure 3 with these new models, which are now consistent.
>
> * Additional Analysis on time-series features.
> - We apologize for not including this analysis in the original submission. We used the linear nu-SVR model to compare the weights of all features, which are normalized before training the SVR. We found the following main insights across datasets:
>
> a. The time-series (TS) features on average have higher weights than HP and AP features (which is confirmed by our ablation studies in Table 2 as well).
> b. The original validation accuracies (y_t) have higher weights on average than the first-order differences (y_t’) and second-order differences (y_t’’).
> c. In general, later epochs in y_t have higher weights.  In other words, the latest performance of the model available for prediction is much more important for performance prediction than initial performance, when the model has just started training. This makes intuitive sense.
> d. Early values of (y_t’) also have high weights, which indicates learning quickly in the beginning of training is a predictor of better final performance in our datasets.
> e. Among AP and HP features, the total number of parameters  and depth are very important features for the CNNs, and  they are assigned weights comparable or higher than late epoch accuracies (y_t). However, they have much lower weight in the LSTM experiment.  The number of filters in each convolutional layer also has a high positive weight for CNNs. In general, architectural features are much more important for CNNs as compared to LSTMs. Hyperparameters like initial learning rate and step size were generally not as important as the architectural features, which is corroborated by the ablation study in Table 1.
>
> We have included these details in Appendix Section F and Figure 10.
>
>
> * “In Table 1 the linear SVM uniformly outperforms the RBF SVM, so why use the RBF version?”
>
> - To ensure consistency across experiments, we have rerun all our results such that SVR now has its kernel as a hyperparameter that is also searched over.
>
> * “how is speedup computed in Figure 4.”
>
> - We compute speedup as (# iterations used without early stopping) / (# of iterations used with early stopping). In Figure 4, we compare the total number of iterations used in a full MetaQNN experiment to a simulation where we early stop models based on the prediction of an SRM (as detailed in Section 4.1. We have included these details in the caption of Figure 4 in the updated version.
>
> * “whether 0.00006 seconds vs 0.024 seconds is something we should care about in this kind of work”
>
> - We do not wish to claim that the difference between 0.00006 seconds vs 0.024 seconds is significant in this context, and we have deemphasised the comparison to prior work on speed in the updated text. However, the difference between the time for one inference required by  our method (0.00006 seconds) and LCE (1 minute) can result in difference in overhead. Since it is necessary to continue training a model on GPU while early stopping prediction is being performed, an overhead equal to the time required to evaluate the early prediction model will be added. If we assume that it takes 1 minute to evaluate LCE, the overhead for the Zoph and Le (2017) experiment---which trains 12800 models---would be (12800*1 min) or 8.8 GPUdays. The equivalent time for our method would be (12800*0.00006 s) or 8*10^-6 GPUdays.
>
>
> * “r-squared as a measure of performance”
>
> - R-square allows us to evaluate the accuracy of a performance prediction model across the search space, which is important because we do not want the performance prediction model to overestimate the performance of poorly-performing architectures nor to underestimate the performance of well-performing architectures.  Having a predictor that works well in all parts of the search space is useful for sequential model based algorithms. For example, in Q-Learning, if you have two poor models, it is useful for the agent to still know which one was relatively better. Finally, Figure 2 qualitatively shows that our models work well across performance ranges.
>
> * Typos / Details
> - We have fixed these typos/errors. We appreciate your careful reading!

---

### Official Review · AnonReviewer2 · 2017-11-28
**Strong performance predictions possible for budgets >> 100 configurations**

**Rating:** 6
**Confidence:** 5

**Review:**

This paper shows a simple method for predicting the performance that neural networks will achieve with a given architecture, hyperparameters, and based on an initial part of the learning curve.
The method assumes that it is possible to first execute 100 evaluations up to the total number of epochs.
From these 100 evaluations (with different hyperparameters / architectures), the final performance y_T is collected. Then, based on an arbitrary prefix of epochs y_{1:t}, a model can be learned to predict y_T.
There are T different models, one for each prefix y_{1:t} of length t. The type of model used is counterintuitive for me; why use a SVR model? Especially since uncertainty estimates are required, a Gaussian process would be the obvious choice.

The predictions in Section 3 appear to be very good, and it is nice to see the ablation study.

Section 4 fails to mention that its use of performance prediction for early stopping follows exactly that of Domhan et al (2015) and that this is not a contribution of this paper; this feels a bit disingenious and should be fixed.
The section should also emphasize that the models discussed in this paper are only applicable for early stopping in cases where the function evaluation budget N is much larger than 100.
The emphasis on the computational demand of 1-3 minutes for LCE seems like a red herring: MetaQNN trained 2700 networks in 100 GPU days, i.e., about 1 network per GPU hour. It trained 20 epochs for the studied case of CIFAR, so 1-3 minutes per epoch on the CPU can be implemented with zero overhead while the network is training on the GPU. Therefore, the following sentence seems sensational without substance: "Therefore, on a full meta-modeling experiment involving thousands of neural network configurations, our method could be faster by several orders of magnitude as compared to LCE based on current implementations."

The experiment on fast Hyperband is very nice at first glance, but the longer I think about it the more questions I have. During the rebuttal I would ask the authors to extend f-Hyperband all the way to the right in Figure 6 (left) and particularly in Figure 6 (right). Especially in Figure 6 (right), the original Hyperband algorithm ends up higher than f-Hyperband. The question this leaves open is whether f-Hyperband would reach the same performance when continued or not.
I would also request the paper not to casually mention the 7x speedup that can be found in the appendix, without quantifying this. This is only possible for a large number of 40 Hyperband iterations, and in the interesting cases of the first few iterations speedups are very small. Also, do the simulated speedup results in the appendix account for potentially stopping a new best configuration, or do they simply count how much computational time is saved, without looking at performance? The latter would of course be extremely misleading and should be fixed. I am looking forward to a clarification in the rebuttal period.
For relating properly to the literatue, the experiment for speeding up Hyperband should also mention previous methods for speeding up Hyperband by a model (I only know one by the authors' reference Klein et al (2017)).

Overall, this paper appears very interesting. The proposed technique has some limitations, but in some settings it seems very useful. I am looking forward to the reply to my questions above; my final score will depend on these.

Typos / Details:
- The range of the coefficient of determination is from 0 to 1. Table 1 probably reports 100 * R^2? Please fix the description.
- I did not see Table 1 referenced in the text.
- Page 3: "more computationally and" -> "more computationally efficient and"
- Page 3: "for performing final" -> "for predicting final"


Points in favor of the paper:
- Simple method
- Good prediction results
- Useful possible applications identified

Points against the paper:
- Methodological advances are limited / unmotivated choice of model
- Limited applicability to settings where >> 100 configurations can be run fully
- Possibly inflated results reported for Hyperband experiment

---

> ### Author Response · Authors · 2018-01-04
> **Responses**
>
> Thank you for your thoughtful review!
>
> * Why use SVR instead of a Gaussian Process?
>
> - One goal of this work was to make early stopping practical to use for arbitrary, albeit large-scale, architecture searches. This is why we constrained ourselves to simple regression models that have many quick and standardized implementations available. In order to train a model on as few as 100 data points, we also needed models that have low sample complexity. At your suggestion, we ran some experiments with Gaussian Processes and Bayesian Linear Regression (BLR). We found Gaussian Processes with a standard RBF covariance kernel  performed very poorly. However, similar to OLS, Bayesian Linear Regression (BLR) performed comparably to SVR in performance prediction (see Figure 3 in the updated text). We also found that a BLR SRM actually achieved a faster speedup on MetaQNN than a SVR SRM; however, the BLR SRM had suboptimal performance when used with f-Hyperband on the SVHN dataset (see Figure 4 and 6 in the updated text). In summary, our observation that simple regression models trained with features based on time-series performance, architecture parameters, and training parameters accurately predict the final performance on networks holds well, irrespective of the regression model used.
>
> * Performance Prediction for Early Stopping in  Domhan et al (2015)
>
> - We did not wish to claim in the text that the use of early stopping for performance prediction is a contribution of our paper. To clarify, we have now included a note in the updated text that our early stopping formulation follows that of Domhan et al (2015).
>
> * Computational cost comparison with LCE
>
> - Thanks for your detailed comment on this. You are correct in that early stopping can be implemented on a CPU while the GPU continues to train the architecture. However, since it is necessary to continue training a model on GPU while early stopping prediction is being performed on the CPU, an overhead equal to the time required to evaluate the early prediction model once will be added. If we assume that it takes 1 minute to evaluate LCE, the overhead for the Zoph and Le (2017) experiment---which trains 12800 models---would be (12800*1 minute) or 8.8 GPUdays. The equivalent time for our method would be (12800*0.00006 seconds) or 8*10^-6 GPUdays. That said, we agree with you that comparing computational cost of different empirical methods is tricky and we have updated the text to de-emphasize the comparison with prior work on speed.  The most important metric to compare methods should be prediction accuracy.
>
> * Clarification Hyperband Experiment
> Responses to questions on the hyperband experiment.
>
> 1. Extending f-Hyperband all the way to the right:
> - In the updated text, we have extended f-Hyperband in both subfigures of Figure 6 as you suggest. When you give f-Hyperband the same number of raw iterations as vanilla Hyperband (i.e. extending Figure 6), f-Hyperband in fact outperforms  vanilla hyperband consistently (with more than the standard error between seeds) on both experiments with both settings for kappa.
>
> 2. Questions about speedup
> - Our apologies for not explaining the claim on speedup in detail. We indeed did not consider difference in performance when claiming 7x speedup. We have now completed experiments where we use pretrained SRMs, and then calculate the speedup and see the difference in performance. We found that the most aggressive settings of early stopping were detrimental to the performance, and we were only able to safely obtain a 4x speedup on CIFAR-10 and a 3x speedup on SVHN, while not compromising on the ability to find the optimal configuration. We have added the results of this experiment to the appendix and updated our claim. Thank you again for pointing this out!
>
> 3. Prior work on speeding up hyperband
> - We have included a reference to the Klein et al. (2017) experiment with hyperband in Section 4.2.  This is the only prior work that we are aware of as well. We did not include a direct comparison to Klein et al. in Figure 6, since f-Hyperband relies on early stopping for speedup, while Klein et al. use BNN as an acquisition function for choosing the next models to evaluate. However, we also included a new experiment in the Appendix (Section D) which uses SVR trained on only architecture features and hyperparameters as an acquisition function in a similar manner to Klein et al. (2017). We found that this did not help much over f-Hyperband.
>
> * Typos / Details
> - We have fixed these typos/errors. We appreciate your careful reading!

---

> > ### Comment · AnonReviewer2 · 2018-01-13
> > **Comments on author response**
> >
> > Thanks for the careful response; it helped clarify several of my questions and also fixed the previously inflated speedup result of 7x at the end of the paper. Extending Figure 6 was also helpful.
> >
> > It is nice to see the new results for BLR. I am confused, though: which features does this use? Why are the results of BLR and OLS are not identical for Table 1 and Figure 2? Since this only depends on means and not uncertainty predictions, they ought to be the same if they used the same features. I am also surprised that Gaussian processes do not work, but that BLR does. BLR is equivalent to a Gaussian process with a particular kernel, so the issue must be the choice of kernel. With very different types of features, a RBF kernel may need an automatic relevant determination (ARD) kernel.
> >
> > I largely agree with the authors' take on the runtime for prediction. I'd like to mention that even 8.8 GPU days overhead would not matter for the Zoph & Le experiments, which required 800 GPUs for two weeks, making an overhead of 8.8 GPU days less than 0.1% percent overhead. But what the authors write in the updated version is perfectly fine, and it is of course very nice to have extremely cheap predictions. This might also enable using these predictions in an inner loop of a more complex method. So, this is clearly a plus of the method.
> >
> > The authors reran all experiments with an automatic choice between a linear and a kernel SVR, and I'm surprised that the results got a lot worse for ResNets/Cuda-ConvNet; compare the numbers for TS+AP+HP in Table 2: R2 values of now 86.05 vs. previously 91.8 or 95.69. Actually, the columns for ResNets & Cuda-ConvNet are now identical, so there must be (at least) a copy-paste error in this table. That, combined with the worsened results, does not leave me very confident about the updated experiments (which, of course, had to be done under time pressure during the rebuttal, increasing the risk of errors).
> >
> > Overall, I have some doubts remaining concerning the inconsistency between OLS and BLR, and the bugs in Table 2. Nevertheless, I would hope that the authors can fix these for the camera ready version; therefore, I still rate this paper just above the acceptance threshold, since I like the method's simplicity and the strong results. If accepted, I encourage the authors to fix Table 2, OLS vs. BLR, and give GPs another shot with an ARD kernel. (If rejected, I'd propose the same for an eventual resubmission.)

---

### Official Review · AnonReviewer1 · 2017-11-30
**Good simple idea, paper hard to read, lack explanations**

**Rating:** 4
**Confidence:** 3

**Review:**

This paper proposes the use of an ensemble of regression SVM models to predict the performance curve of deep neural networks. This can be used to determine which model should be trained (further). The authors compare their method, named Sequential Regression Models (SRM) in the paper, to previously proposed methods such as BNN, LCE and LastSeenValue and claim that their method has higher accuracy and less time complexity than the others. They also use SRM in combination with a neural network meta-modeling method and a hyperparameter optimization one and show that it can decrease the running time in these approaches to find the optimized parameters.

Pros: The paper is proposing a simple yet effective method to predict accuracy. Using SVM for regression in order to do accuracy curve prediction was for me an obvious approach, I was surprised to see that no one has attempted this before. Using features sur as time-series (TS), Architecture Parameters (AP) and Hyperparameters (HP) is appropriate, and the study of the effect of these features on the performance has some value. Joining SRM with MetaQNN is interesting as the method is a computation hog that can benefit from such refinement. The overall structure of the paper is appropriate. The literature review seems to cover and categorize well the field.

Cons: I found the paper difficult to read. In particular, the SRM method, which is the core of the paper, is not described properly, I am not able to make sense of the description provided in Sec. 3.1. The paper is not talking about the weaknesses of the method at all. The practicability of the method can be controversial, the number of attempts require to build the (meta-)training set of runs can be huge and lead to something that would be much more costful that letting the runs going on for more iterations.

Questions:
1. The approach of sequential regression SVM is not explained properly. Nothing was given about the combination weights of the method. How is the ensemble of (1-T) training models trained to predict the f(T)?
2.  SRM needs to gather training samples which are 100 accuracy curves for T-1 epochs. This is the big challenge of SRM because training different variations of a deep neural networks to T-1 epochs can be a very time consuming process. Therefore, SRM has huge preparing training dataset time complexity that is not mentioned in the paper. The other methods use only the first epochs of considered deep neural network to guess about its curve shape for epoch T. These methods are time consuming in prediction time. The authors compare only the prediction time of SRM with them which is really fast. By the way still, SRM is interesting method if it can be trained once and then be used for different datasets without retraining. Authors should show these results for SRM.
3. Discussing about the robustness of SRM for different depth is interesting and I suggest to prepare more results to show the robustness of SRM to violation of different hyperparameters.
4. There is no report of results on huge datasets like big Imagenet which takes a lot of time for deep training and we need automatic advance stopping algorithms to tune the hyper parameters of our model on it.
5. In Table 2 and Figure 3 the results are reported with percentage of using the learning curve. To be more informative they should be reported by number of epochs, in addition or not to percentage.
6. In section 4, the authors talk about estimating the model uncertainty in the stopping point and propose a way to estimate it. But we cannot find any experimental results that is related to the effectiveness of proposed method and considered assumptions.

There are also some  typos. In section 3.3 part Ablation Study on Features Sets, line 5, the sentence should be “Ap are more important than HP”.

---

> ### Author Response · Authors · 2018-01-04
> **Responses**
>
> Thank you for your thoughtful review and suggestions! We agree with you that predicting final performance of neural networks using a regression model trained with features based on time-series accuracies, architecture parameters, and training parameters is a surprisingly simple and effective idea. We hope that this work is published to advance the literature in this previously under-explored area, and to establish proper, simple baselines in future work in this field. We apologize if the exposition was not clear and we have included detailed explanation of our method below and also in the updated text.
>
> In response to your specific questions:
>
> 1. “How is the ensemble of (1-T) training models trained to predict the f(T)?”
>
> - We train T-1 separate regression models, where each sequential model uses one more point of the validation curve (i.e. the k’th model would use validation measurements from the first k epochs). We do not ensemble these models for prediction, so there are no combination weights. For early stopping, if we have trained a model to k epochs, then we use the k’th regression model to compute a performance estimate.
>
> 2.1 Comparing time complexity of training and inference of SRMs and other methods
>
> - While SRMs do require training 100 configurations to build a prediction model, this does not add any extra computational overhead in the large-scale architecture searches---which train several thousand models per experiment, each for a large number of epochs. For example,  Zoph and Le (2017) trained 12,800 models. Moreover, as experiments in Figure 4 show, even after taking the time to train a 100 models into account, SRMs are significantly faster than previous methods (e.g., LCE) that do not require a meta-training set. Similarly, our method obtains speedup on Hyperband, because, again, there is no overhead from incorporating our method into any search method. In sum, for appropriate applications like hyperparameter search and neural network architecture search, the computational expense of our method should not be a hindrance especially since most searches involve training models on a GPU, and we train our SRMs on CPU.
>
> 2.2  Using SRMs on different datasets without retraining
>
> - Since we used quite different architecture types (ResNets, LSTMs, basic CNNs) and hyperparameter sets (e.g. stepwise exponential learning rate decay) in individual experiments to demonstrate the versatility of our method, the learning curves across datasets were too dissimilar for transfer learning to work well. However, this is certainly an important area of future research with useful applications.
>
>
> 3. “More results to show the robustness of SRM to violation of different hyperparameters.”
>
> -  According to your suggestion, we ran several additional experiments in this vein, where we trained an SRM with models with hyperparameter values below/above the median and tested their performance on the remainder of the models. SRMs generally showed consistent performance across such splits. For example, an SRM trained using LSTMs with # layers below the median obtained an r-squared of 0.967 on the remainder and the performance for the other split was 0.986. We have included results of several such experiments in Appendix Section H.
>
> 4. No results on huge datasets like big Imagenet
>
> -  Unfortunately we do not have enough resources to experiment with this method on big Imagenet. We hope that our work inspires such investigations, especially in the industry setting, where our method can be valuable.
>
> 5. “Results In Table 2 and Figure 3 should be reported by number of epochs, in addition or not to percentage.”
>
> - We report the results in terms of percentage for easy visual comparison across datasets within the figure. In the camera-ready version, we will replicate this figure in the Appendix using number of epochs. The total number of epochs for each experiment are found in Section 3.2.
>
> 6. “Experimental results on estimating the model uncertainty that are related to the effectiveness of proposed method and considered assumptions.”
>
> - We conducted more analysis on our considered assumptions (Appendix Section E).  We test the assumption on Gaussian-distributed errors using examples of the held out set error distributions compared with the Gaussian computed from training set errors. This assumption holds reasonably well (Appendix Figure 8). Figure 9 shows plots of the mean log likelihood of the held out errors being drawn from the Gaussian parameterized by the mean and variance of the training errors. These plots show that the log likelihood is very close to the baseline (mean log likelihood of samples drawn from the same Gaussian), which also shows that the assumption holds well. Finally, the effectiveness of the proposed method is also illustrated by results in Figure 4, where our algorithm successfully recovers the optimal model in most cases.

---

### Author Response · Authors · 2018-01-04
**Summary of responses to reviewers and paper updates**

Thanks to all of our reviewers for their thoughtful comments. We have incorporated many if not all of their suggestions into our updated text. We have added much additional analysis of the method into the appendix and hopefully clarified and improved the text. We summarize our additional analysis below.

List of additional analyses performed:

a. Because there was no clear winner between SVR kernels (linear versus RBF), we included the kernel into the hyperparameter search space such that each model in the SRM chooses the best kernel dynamically. We have updated all results, both in performance prediction and early stopping experiments, to reflect the new SVR SRM.

b. We have added results for Bayesian Linear Regression (BLR). We include performance prediction results for BLR in Table 1 and Figure 3. We also include results using a BLR SRM (using natural uncertainty estimates instead of ensemble estimates) and find that in some cases it outperforms the SVR SRM (see Figures 4 and 6). We believe this further strengthens our main point that simple models can provide accurate neural network prediction performance.

c. Appendix Section D: We have included results where we use a SVR model trained on only architecture features and hyperparameters (no time-series) as an acquisition function used to choose configurations to evaluate within f-Hyperband (Similar to Klien et al. 2017). We did not find any significant improvement from adding this acquisition function.

d. Appendix Section E: We include analysis of the Gaussian error assumption used to estimate uncertainty in frequentist SRMs. We empirically found that the assumption holds well by comparing the held out error distributions to training error distributions.

e. Appendix Section F: We expound upon the ablative analysis presented in Table 2 to give more intuition for which features are useful in predicting future performance through analyzing the weights of linear performance prediction models.
We added Figure 13 to Appendix Section G, which is complementary to Figure 12. In this experiment we show the potential speedup if one uses pretrained SRMs with f-Hyperband.

f. Appendix Section H: We add more results on the robustness of SRMs to out-of-distribution configurations (we originally just included one such experiment in the main text).

---

### Decision · Program_Chairs · 2018-01-29
**ICLR 2018 Conference Acceptance Decision**

**Decision:**

Invite to Workshop Track

**Comment:**

The paper proposes to use simple regression models for predicting the accuracy of a neural network based on its initial training curve, architecture, and hyper-parameters; this can be used for speeding up architecture search. While this is an interesting direction and the presented experiments look quite encouraging, the paper would benefit from more evaluation, as suggested by reviewers, especially within state-of-the-art architecture search frameworks and/or large datasets.